# Microwave-Assisted Glycerol Etherification Over Sulfonic Acid Catalysts

**DOI:** 10.3390/ma13071584

**Published:** 2020-03-30

**Authors:** Laura Aguado-Deblas, Rafael Estevez, Marco Russo, Valeria La Parola, Felipa M. Bautista, Maria Luisa Testa

**Affiliations:** 1Departamento de Química Orgánica, Universidad de Córdoba, Campus de Rabanales, Ed. Marie Curie, 14014 Córdoba, Spain; aguadolaura8@gmail.com (L.A.-D.); q72estor@uco.es (R.E.); qo1baruf@uco.es (F.M.B.); 2Istituto per lo Studio dei Materiali Nanostrutturati, ISMN-CNR, Via Ugo La Malfa 153, 90146 Palermo, Italy; marco.russo@ismn.cnr.it (M.R.); valeria.laparola@cnr.it (V.L.P.)

**Keywords:** glycerol, etherification, solid acid catalyst, microwave

## Abstract

Glycerol is the main by-product of biodiesel production. For this reason, its valorization into value-added products, by using green procedures, represents an important goal. Different sulfonic acid silica- or titania-based catalysts were prepared, characterized and tested in the glycerol etherification process, assisted by microwaves, in order to obtain biodiesel additives. The surface and structural properties of the catalysts were investigated by means of N_2_ adsorption isotherms, thermogravimetric analysis (TGA), X-ray photoelectron spectroscopy (XPS) and acid capacity measurements by X-Ray Fluorescence Spectroscopy (XRF). The best performance in terms of activity was achieved in the presence of the sulfonic function directly linked to the amorphous silica. By the correlation of the structure properties of the materials and their activity, the performance of the catalysts was shown to be influenced mainly by the surface area, pore volume and acidity. Recycling experiments performed over the most active systems showed that the sulfonic silica-based materials maintained their performance during several cycles.

## 1. Introduction

The use of raw materials such as biomass for the production of fuels and chemicals is a sustainable alternative to fossil sources. In particular, the triglyceride part of biomass is principally used for biodiesel production through the well-known transesterification of vegetable oils with methanol to yield Fatty Acid Methyl Esters (FAMEs).

Biodiesel presents the intrinsic advantages of degradability, low toxicity and renewability, but, during its production, glycerol (G) is also obtained as the principle by-product (10% w/w). Glycerol has to be completely removed from the mixture because at the high temperatures reached in engines, it promotes the formation of acrolein and its polymers which are deleterious for the engine as well as being polluting. The process to separate glycerol is still expensive and tedious, and for this reason, it still represents the major drawback of biodiesel production. 

Different strategies are being studied by the research community to make the process more sustainable and cost-effective. On the one hand, different alternative methods are being investigated in order to convert vegetable oils into high quality diesel fuels avoiding the formation of glycerol [1]. Luna et al. studied the partial transesterification of triglycerides, generating only two equivalent molecules of FAMEs, maintaining the third equivalent of fatty acids in the form of monoglyceride (MG) [2,3]. In this line, Testa et al. studied one-pot, microwave-assisted catalytic transformation of vegetable oil into glycerol-free biodiesel [4]. On the other hand, glycerol can be considered one of the most important platform molecules, so its valorization to obtain value-added products is being investigated, focusing on hydrogenolysis, dehydration, esterification, etherification and acetalization reactions [5,6,7].

The etherification of glycerol with tert-butyl alcohol (TBA) to produce an oxygenated additive for fuel has become one of the best options to valorize glycerol in the biodiesel industry. The reaction, described in Figure 1, makes it possible to obtain mono-tert-butyl glycerol ethers (MTBGs), di-tert-butyl-glycerol ethers (DTBGs) and tri-tert-butyl glycerol ether (TTBG). 

While MTBGs cannot be blended with fossil fuel, the DTBGs and TTBG, i.e., the so-called high ethers (h-GTBE), can be employed as diesel and biodiesel additives [8,9] to improve cold properties and reduce the consumption and gas emissions [10,11].

Behr et al. studied the reaction in homogenous conditions by the use of p-toluenesulfonic acid or mineral acids, e.g., sulfuric acid [12]. In order to make the process “greener”, the use of the heterogeneous catalytic materials is preferred. In fact, the use of a solid catalyst is indeed very favorable from a sustainable point of view, due to its simple separation and regeneration, as well as the recycling potential of the material for different cycles. This implies a high level of efficiency of the materials accompanied by a decrease in the processing cost. Several systems have been studied and, in general, the best catalytic results were obtained using Brønsted acid catalysts, including ion exchange resins [13,14], zeolites [15] and sulfonic-functionalized materials (zeolites [16], silicas [17], silicoaluminophosphates [9] and sulfonated black carbons [18]). Several studies have highlighted the importance of the acidity of the materials in their catalytic performance. Estevez et al. [19] found a direct correlation between acidity and the h-GTBE yield, both over amorphous organosilicas-aluminumphosphates [9] and over sulfonic hybrid silica. The correlation was also confirmed by Galhardo et al. over different carbon-based catalysts functionalized with sulfonic groups by different acid treatments [20].

Along with the acidity, the textural properties of the materials also play an important role in their catalytic activity. In general, a better accessibility of the reactant to the active sites promotes a higher catalytic yield of the desired products [15,19,21,22]. This implies the presence of an acidic core on the surface catalysts, or inside the pores that are big enough to allow the diffusion of the reactants and/or products to occur.

The hydrophilic/hydrophobic properties of the materials also have to be taken into account, not only for the activity, but also for the deactivation that the catalysts suffer during the reaction. Gonçalves et al. [23] studied the etherification reaction over sulfonated carbons obtained from agroindustrial wastes; they concluded that the hydrophilic character of the carbons made possible the absorption of the water generated during the reaction, palliating the negative influence that this water usually has towards the catalytic activity and the selectivity to h-GTBE. Analogous results were obtained over silica-derived catalysts with sulfonic acid groups in the structure [9,19]. The silanol groups of the structure would avoid the problem of the deactivation of the acid sites in the catalysts in the initial steps.

On these premises and as a continuation of our interest in the solid acid catalysts [24] applied to glycerol valorization for the production of added value products [25], in the present work, different sulfonic materials, both silica- and titania-based, were tested on glycerol etherification of glycerol with tert-butyl alcohol (Figure 1). Following our previous studies [9,16] with the goal of making the process greener and more cost-effective, the reaction was carried out using microwaves. 

In particular, synthesized amorphous silica and titania were functionalized with a propylsulfonic group in samples **1** and **4** respectively. Sulfonic silica catalysts were doped with 10% w/w Ti, (**2** and **3**) in order to evaluate the effect of Ti on the properties of the material and, as consequence, on the catalytic performance. Moreover, the effect of alkyl chains on catalytic activity was considered both in the cases of silica and titania materials by the functionalization of the sulfonic group directly linked by a grafting procedure (samples **5** and **6,** respectively). For the sake of comparison, the procedures for the synthesis of sulfonic catalysts are described in Figure 2. 

Some of the considered materials were already used for acid catalyzed reactions such as glycerol acetylation [25], fructose dehydration [26] or transesterification [24], showing high catalytic activity related to the characteristic of the materials. Due to their properties and in order to increase the applications, these materials were tested on microwave-assisted glycerol etherification. The procedure was optimized by changing the temperature and the reaction time, whereas the catalyst amount and the molar ratio TBA/G had already been optimized in previous studies [9,19] over different catalysts, indicating in all cases that the best conditions were 5% w/w of catalysts and a 4:1 TBA:G molar ratio. Recycling experiments were performed with the most active systems. The materials were characterized by N_2_ adsorption isotherms, thermo-gravimetric analysis (TGA), X-ray photoelectron spectroscopy (XPS) and acid capacity measurements by X-Ray Fluorescence Spectroscopy (XRF). These characterizations allowed us to correlate the physical and chemical properties of materials with their catalytic activity. 

## 2. Materials and Methods 

### 2.1. Preparation of Catalysts 

#### 2.1.1. Synthesis of the Propyl SO_3_H Amorphous SiO_2_ as It (**1**) or 10% Ti Doped (**2**)

Propyl SO_3_H silica was synthesized by the sol–gel technique according to a published procedure [24]. Tetraethoxysilane (TEOS, 7.5 mL, 0.034 mol) was dissolved in ethanol (5 mL) and stirred at 45 °C for 15 min. Then, in the case of catalyst 2, 5 mL aqueous solution of acetic acid with pH 5 was added to the mixture together with 10% w/w of Ti, as Ti(iPrO)_4_. This was followed by the addition of 10% mol/mol of 3-mercaptopropyltrimethoxy silane (MPTMS) and hydrogen peroxide (35% w/v solid:liquid ratio of 1:18). The solution was heated to 80 °C and left at this temperature until the formation of the gel occurred. The obtained wet gel was dried at 110 °C overnight.

#### 2.1.2. Synthesis of 10% Ti Doped Propyl SO_3_H Amorphous SiO_2_ (**3**)

For the synthesis of catalyst 3, three steps were performed. TEOS (30 mL, 0.135 mol) was dissolved in ethanol (20 mL) and stirred at 45°C for 15 minutes. Then, 19 mL of aqueous solution of acetic acid at pH 5 was added to the mixture followed by 10% w/w of Ti as Ti(iPrO)_4_. The temperature was increased to 80 °C until the formation of the gel occurred. The obtained wet gel was dried at 110 °C overnight and then calcined in air at 450 °C/ 4 h (ramp of 5 °C/min). For the introduction of a propyl sulfonic group, a grafting procedure was used. In a typical procedure, a mixture of 2.00 g of calcined TiSiO_2_ in 35 mL of dry toluene and MPTMS (3.33 mmol) was refluxed for 24 h at 110 °C. The materials were then recovered by filtration, washed several times with toluene and ethanol and dried at 120 °C overnight. Thereafter, the –SH groups were oxidized to sulfonic groups with hydrogen peroxide (35% w/v solid:liquid ratio of 1:18) at room temperature for 24 h. After filtration, the solid was dried at 120 °C overnight.

#### 2.1.3. Synthesis of SiO_2_ Support

To obtain amorphous silica, TEOS (30 mL, 0.135 mol), was dissolved in ethanol (20 mL) and stirred at 45 °C for 15 min. Then, 19 mL aqueous solution of acetic acid with pH 5 was added to the mixture. The temperature was increased to 80 °C until the formation of the gel occurred. The obtained wet gel was dried at 110 °C overnight and then calcined in air at 450 °C/ 4 h (ramp of 5 °C/min).

#### 2.1.4. Synthesis of TiO_2_ Support

Mesostructured titania was prepared according to a previous published procedure [27], using a nonionic amphiphilic triblock polymer, Pluronic P123, as a template. The polymer was dissolved in 2-propanol containing HCl diluted in water. The mixture was stirred overnight at 35 °C in a 250 mL one neck flask. Ti(i-PrO)_4_ was quickly added to this solution and stirred for 24 h at the same temperature. The molar composition was 1.0 Ti(i-PrO)_4_ : 34 C_3_H_7_O : 0.04 HCl : 3 H_2_O : 0.02 P123. The milky suspension was aged at 40 °C for 24 h in a closed polypropylene bottle. The solid product was filtered, washed with water and ethanol and calcined in air at 500 °C for 5 h (ramp of 2 °C/min). 

#### 2.1.5. Synthesis of Propyl Sulfonic Acid Titania (**4**)

In a typical synthesis by grafting, a mixture of titanium dioxide (2.00 g) in dry toluene (35 mL) and MPTMS (3.33 mmol) was refluxed for 24 h at 110 °C. The material was then recovered by filtration, washed with toluene and ethanol, and dried at 120 °C overnight. Thereafter, the mercaptopropyl groups were oxidized to sulfonic groups with 35% hydrogen peroxide solution (2 mL) in methanol (20 mL) at room temperature for 24 h. After filtration, the solid was dried at 80 °C overnight.

#### 2.1.6. Synthesis of Sulfonic Acid Titania (**5**) or Silica (**6**)

In a typical synthetic procedure, chlorosulfonic acid (0.25 mL, 3.6 mmol) was added dropwise to a suspension of TiO_2_ or SiO_2_ (1.0 g, 1.2 mmol) in dry dichloromethane (10 mL) at 0 °C with stirring for 2 h until the HCl gas evolution stopped. The temperature rose to room temperature and the mixture was stirred for an additional 2 h. Then, the mixture was filtered and washed with ethanol and dried at room temperature to obtain either TiO_2_–SO_3_H (**5**) or SiO_2_–SO_3_H (**6**).

### 2.2. Catalysts Characterization

The textural properties were obtained using a Micromeritics ASAP2020 Plus 1.03 (Micromeritics, Ottawa, Canada). The fully computerized analysis of the N_2_ adsorption isotherm at 77 K allowed us to obtain, through the BET method in the standard pressure range 0.05–0.3 p/p_0_, the specific surface areas of the samples. The total pore volume, Vp, was evaluated on the basis of the amount of nitrogen adsorbed at a relative pressure of 0.998, while mesopore size distribution values and mesopore volumes were calculated by applying the BJH model in the range of p/p_0_ of 0.1–0.98.

The thermogravimetric analyses of the samples were performed in air using the TGA 1 Star System of Mettler Toledo (Mettler Toledo, Schwerzenbach, Switzerland). About 10 mg of sample was heated from room temperature to 100 °C, left at this temperature for 30 min and then heated to 1000 °C at a rate of 10 °C/min in 40 mL/min of air.

X-ray photoelectron spectroscopy analyses were performed with a VGMicrotech ESCA 3000 Multilab (VG Scientific, Sussex, UK), equipped with a dual Mg/Al anode. The spectra were excited by the unmonochromatized Al_K source (1486.6 eV) run at 14 kV and 15 mA. The analyzer operated in constant analyzer energy (CAE) mode. For the individual peak energy regions, a pass energy of 20 eV set across the hemispheres was used. Survey spectra were measured at 50 eV pass energy. The sample powders were analyzed as pellets mounted on a double-sided adhesive tape. The pressure in the analysis chamber was of the order of 10^−8^ Torr during data collection. The constant charging of the samples was removed by referencing all the energies to the C 1s set at 285.1 eV arising from the adventitious carbon. The invariance of the peak shapes and widths at the beginning and end of the analyses ensured the absence of differential charging. Analyses of the peaks were performed with the CasaXPS software Version 2.3.18PR1.0. Atomic concentrations were calculated from peak intensity using the sensitivity factors provided with the software. The binding energy values are quoted with a precision of ± 0.15 eV and the atomic percentage with a precision of ± 10%.

The sulfur content was also determined by X-ray fluorescence measurements on a ZSX primus IV instrument from Rigaku (Tokyo, Japan), using fundamental parameters. The acid capacity of the solids was determined from the amount of sulfur determined by XRF, given that all the sulfur in the samples was in sulfonic form, as observed from the XPS spectra of the solids.

### 2.3. Catalytic Reaction

Microwave experiments were carried out in a CEM-DISCOVER (Mathews, CA, USA) apparatus with PC control under the reaction conditions described elsewhere [9], at which the diffusion limitations were ruled out. The microwave power increased until the desired temperature, previously set, was attained (measured by an infrared probe). Then, the power oscillated in order to keep the reaction temperature at the desired value. The maximum value of the microwave was 300 W. In a typical run, the composition of the reaction mixture was 0.4 g glycerol, a TBA/G molar ratio of 4 and constant catalyst loading of 5 wt.% (referred to initial glycerol mass). The total volume of the reactant mixture was 2 mL. When the reaction finished, the sample was cooled in an ice bath, filtered off and subsequently analyzed.

Blank experiments using either microwave or conventional heating showed that the mixture of TBA/G did not react in the absence of the catalyst under the experimental conditions employed. 

Samples were analyzed by gas chromatography (GC) using a Hewlett Packard 5890 series II (Santa Clara, CA, USA), equipped with a Supelcowax 10 capillary column, and a flame ionization detector (FID), using 4-chlorotoluene as an internal standard [9,19]. The analysis program was 60 °C for 6 min heating at 20 °C/min until a temperature of 240 °C was reached, with an analysis time of 25 min. The conversion of glycerol (X_G_) and product selectivity (S) were determined by the following equations: (1)$$X_{G}\;\left(\%\right)=\frac{\mathrm{mmol}\;\mathrm{of}\;\mathrm{produced}\;\mathrm{TBGs}}{\mathrm{starting}\;\mathrm{mmol}\;\mathrm{of}\;G}\times 100$$
(2)$$S_{i}\;\left(\%\right)=\frac{\left(\mathrm{mmol}\;\mathrm{of}\;\mathrm{compound}\;i\right)}{\left(\mathrm{mmol}\;\mathrm{of}\;\mathrm{produced}\;\mathrm{TBGs}\right)}\times 100$$

It should be noted that diisobutylene (DIB) or secondary products coming from glycerol were not obtained under these conditions. Furthermore, in order to check the reliability of the quantification procedure, at the end of some experiments, the DTBGs and TTBG products, which were not commercially available, were isolated from the final product by column chromatography (1:9 ethyl acetate/hexane). The results obtained (X_G_ and S_i_) were slightly lower (<8%) than those obtained by the gas chromatography analysis.

At the end of the reaction, the catalyst was recovered, washed with ethanol and dried at 100 °C for 24 h. Recycling experiments were performed over the most active catalysts. In this case, the dried spent material was tested again in subsequent runs over four cycles.

## 3. Results and Discussion

### 3.1. Characterization of Catalysts 

As described in Figure 2, two different types of sulfonic materials were prepared: silica- and titania-based catalysts. 

Some of the catalysts have already been synthetized, well characterized and tested for other reactions. In particular, sulfonic silica-based material (**1**) was used in a transesterification reaction of short chain esters [24], while titania-based materials (**4,5**) were recently evaluated in fructose dehydration [26].

Table 1 summarizes the results of analysis characterization of the considered catalysts. Catalysts **1** and **2** were synthesised using a one-pot method in which the material was functionalized during its formation. Sulfonic silica catalyst **1** presented a high surface area (616 m^2^g^−1^) along with a high pore volume (0.34 cm^3^g^−1^). The insertion of Ti into the structure, by using the same synthetic procedure (one-pot), promoted a significant decrease in surface area (until 208 m^2^g^−1^), as well as in pore volume (0.02 cm^3^g^−1^). At the same time, changing the synthetic procedure with the use of the grafting method, the results showed a material (**3**) with a high surface area, i.e., 501 m^2^g^−1^, and a pore volume of 0.18 cm^3^g^−1^. In this case, catalyst **3** was synthesized in two steps: the insertion of Ti took place during the sol-gel synthesis, yielding a TiSiO_2_ support material with a high surface area (746 m^2^g^−1^) that, as expected, decreased (501 m^2^g^−1^) after the grafting procedure in order to introduce the propyl sulfonic group. The different values between materials **2** and **3** may be related to the one-pot procedure used for catalyst **2**, in which the starting presence of Ti(iOPr)_4_ and H_2_O_2_ led to the formation of a protonated peroxotitanate complex which gave the material a characteristic yellow-orange colouring [28,29]. The presence of a protonated peroxotitanate complex can affect silica condensation, giving rise to a more packed structure characterized by low pore volume and surface area. This is supported by the studies of Chang et al. on the synthesis of TiO_2_ in the presence of H_2_O_2_. [30] In fact, a progressive shift was observed from anatase to rutile, as was a consequent surface area reduction, related to an increase in the H_2_O_2_/Ti(iOPr)_4_ ratio. In order to study the influence of the organic chain, amorphous silica (SSA 540 m^2^g^−1^ and Vp 0.44 cm^3^g^−1^) was direcly grafted with a sulfonic group, yielding catalyst **6** with a lower surface area, i.e., 406 m^2^g^−1^, and pore volume, i.e., 0.36 cm^3^g^−1^, due to the used procedure. The titania-based catalysts with grafted propyl sulfonic (**4**) or sulfonic group (**5**) have been characterized elsewhere [26]. Starting from a titania support (SSA 45 m^2^g^−1^ and Vp 0.10 cm^3^g^−1^), grafted materials **4** and **5** presented very low surface areas (41 and 37 m^2^g^−1^, respectively) and pore volumes (0.09 and 0.07 cm^3^g^−1^, respectively). In neither case did the grafting procedure influence the textural structure of the support that was maintained.

The incorporation of sulfur in the different materials has been corroborated by different techniques. Figure 3 shows the thermogravimetric analysis (Table 1 and Figure 3) in a temperature range of 100–1000 °C.

Different weight losses can be observed. The first one, under 200 °C, can be attributed to the removal of physisorbed water. The weight loss associated to the loss of (organo)sulfonic acid groups is in the range 300–600 °C. This weight loss is, therefore, indicative of the sulfonic groups present in the materials. The high surface area of silica-based materials promote the link with a higher number of sulfonic groups, above all during the one-pot procedure (15.5 and 20.6 wt.% in the case of propylsulfonic silica as it **1** or Ti doped **2**, respectively). When the organic chain is grafted on the surface after the formation of titania-doped silica, the functionalization of catalyst **3** results in 12.8 wt.%. The direct grafting of the sulfonic groups onto the amorphous silica allowed us to obtain a yield of 6.0 wt.% in the solid **6**. Concerning titania-based materials, in both cases, the loading of sulfonic groups was very low, which may be related to the low surface area (profiles not shown).

Regarding the acidity calculated from XRF, it was similar for the silica-based materials synthesized by the one-pot method, either with or without Ti, 1 and 0.9 mmol H^+^/g (solids **1** and **2**, respectively); see Table 1. By using a grafting method in the Ti silica support, i.e., the Ti SiO_2_ PrSO_3_H catalyst (**3**), a lower amount of sulfur was incorporated (0.7 mmol H^+^/g). When the incorporation of sulfonic groups was made with chlorosulfonic acid on silica (SiO_2_–SO_3_H **6**), a higher amount of sulfur was grafted, creating an acidity of 0.9 mmol H^+^/g. Furthermore, it is noteworthy that the incorporation of sulfur was considerably low (0.3 and 0.2 mmol H^+^/g) for the catalytsts prepared by grafting over a titania support (solids **4** and **5**, respectively) in comparison to those which contained silica. Therefore, the higher functionalization in silica-based materials could be attributed to their high surface area and to the major presence of –OH groups with respect to titanium-based materials. 

An XPS analysis confirmed the efficiency of the oxidation procedure. The S2p peak at 170 eV is typical of sulfonated materials, and nonresidual peaks at 164 eV (typical of thiolated compound) are evidenced (Figure 4).

The S:Si and S:Ti ratios are shown in Table 1. The trend of the ratio is in accordance with the acidity measured by titration analysis, while if we compare this with the thermogravimetric analysis, we find a higher surface loading for catalyst **1.** This discrepancy is probably due to the higher surface area of **1** which allowed better dispersion to occur of a lower number of acidic groups.

### 3.2. Catalytic Activity in the Etherification Reaction

The well-known reaction mechanism of the etherification of glycerol with tert-butyl alcohol is described in Figure 1. In summary, it occurs via a fast protonation of TBA on acid sites, giving rise to a tertiary carbocation that reacts with glycerol, generating MTBGs. The process is sequential; MTBGs react with TBA to form the DTBGs, which react again with TBA to achieve TTBG. Moreover, during the process, water is also produced as a by-product [31,32]. It is important to notice that in the present study, a small amount of isobutylene (IB) <5%, coming from the dehydration of the alcohol, was also observed.

#### 3.2.1. Optimization of Reaction Condition 

Different reaction parameters were studied in order to optimize the process. According to previous research [9,19], the amount of catalysts (5 wt.% referred to initial glycerol mass) and the 4:1 TBA:G molar ratio had already been selected. For this reason, in this study, the reaction temperature and time were evaluated in order to obtain the highest yield of h-GTBE.

The influence of the reaction temperature on glycerol conversion was studied for 15 min using sulfonic propyl silica **1**; the results are described in Figure 5. 

Since the etherification reaction is an endotermic reaction, as expected, glycerol conversion increased with the increase of temperature, reaching the highest conversion value (86%) at 130 °C. However, at a higher temperature (140 °C), the glycerol conversion was lower, i.e., 64% and at the same temperature, a higher amount of isobutylene (9%) and a small amount of diisobutylene (DIB) (3%) were also obtained. The decrease of glycerol conversion at high temperatures was previously observed by Galhardo et al. [20]; this was attributed to the reversible reactions that take place at high temperatures, promoting the thermal decomposition or recombination of intermediates. Likewise, higher temperatures, together with the acidity of the catalysts, promote the formation of IB and the subsequent polymerization into DIB and derivatives as by products, avoiding the production of the desired compounds [19].

To elucidate the factors that affect the reaction to a greater extent (reversible reactions and/or polymerization of IB), the effect of the reaction time at two different temperatures, 130 and 140 °C, was studied (Figure 6). 

When the reaction was carried out at 130 °C, glycerol conversion increased with time up to 15 min, at which point a plateau was almost reached; in fact, the conversion continues slowly, increasing up to 93% at 30 min. A different profile was obtained when the reaction was carried out at 140 °C. In fact, glycerol conversion reached the maximum value (81%) after 10 min and then decreased to 58% at 30 min. This could be ascribed to the hydrolysis of the ether bonds formed during the reaction, and consequently, a conversion of the products into reactants. These results are in agreement with those of a study reported by Beatrice et al. [33] in which an increase of the reaction temperature promoted the thermal decomposition or recombination of reaction intermediates, changing the glycerol conversion and the selectivity. Furthermore, if the cause of the decrease in the glycerol conversion was the formation of IB or DIB, a stabilization in the conversion values would be obtained, but not a decrease thereof.

Concerning the selectivity towards the formation of h-GTBE, a similar trend as that observed for glycerol conversion was obtained at the different temperatures studied (Figure 7). 

Thus, no differences in the S_h-GTBE_ were obtained after 10 min, whereas at 15 and 30 min, the selectivity to h-GTBE was much lower at 140 °C (14% and 12%) with respect to the value of 21% and 29% achieved at the lower temperature. According to the obtained results, the screening of the sulfonic materials was carried out at the chosen temperature, 130 °C.

#### 3.2.2. Screening of the Sulfonic Silica- or Titania-Based Catalysts

Table 2 shows the catalytic performance of sulfonic catalysts (**1–6**) in the microwave-assisted etherification reaction of glycerol with tert-butyl alcohol with a 10-min reaction time. 

As described, when the sulfonic propyl silica catalyst **1** was used, a glycerol conversion of 76% was reached along with 24% of the desired product, h-GTBE. The introduction of Ti into the structure will not improve the performance of the materials in the cases of catalysts **2** and **3**. In particular, when the propyl sulfonic groups were grafted onto Ti-silica (solid **3**), a much better glycerol conversion (64%) and h-GTBE selectivity (14%) was found with respect to the corresponding solid **2**. Moreover, the influence of the organic chain in the performance of the catalysts could be considered by comparing silica-based materials **1** and **6**. Both catalysts presented almost the same acidity, very high activity and analogous performances, as shown in Table 2. This demonstrates that the presence of an alkyl group did not affect the performance of the catalyst. The effectiveness of the silica support with respect to titania is evident by the comparison of catalysts **1** and **4**, which presented the same catalytic site even though the intrinsic material characteristics (such textural properties and acidity) were quite different. Moreover, due to a higher acidity of solid **5** compared to **4**, the sulfonic group direct linked onto the surface of the support led to better performance in terms of both conversion and selectivity. 

The catalysts were also evaluated with a 30-min reaction time when the equilibrium was reached; the data are presented in Table 2 (in brackets). In general, both glycerol conversion and the h-GTBE selectivity increased with the reaction time, maintaining the same performance among the studied materials. 

To sum up, the SiO_2_-PrSO_3_H (**1**) and SiO_2_-SO_3_H (**6**) catalysts were the most active materials, exhibiting the highest values of conversion and yield to h-GTBE. 

In order to verify the influence of the material acidity on the catalytic performance, the yield to h-GTBE vs the acidity of the catalysts was plotted (Figure 8).

In general, a linear tendency is observed, indicating that acidity strongly influences the catalytic reaction, as previously observed by several authors [34,35,36]. However, an exception was observed for catalyst (**2**), i.e., Ti SiO_2_ PrSO_3_H. The one pot Ti SiO_2_ PrSO_3_H catalyst (**2**) exhibited strong acidity (0.9 mmol H^+^/g), so a high value of glycerol conversion and yield to h-GTBE was expected. Nevertheless, the lowest value of glycerol conversion (14%) and yield to h-GTBE (1%) was obtained with this catalyst. 

To explain this behavior, the catalytic performance was also related to the textural properties of the materials. Catalyst **2** exhibited a medium surface area (208 m^2^/g) but a very low pore volume (0.02 cm^3^/g). Therefore, the low pore volume of the catalyst would inhibit the diffusion of the molecules through the pores, avoiding their contact with the active sites of the catalyst and explaining the poor activity of this material. In contrast, for the other catalysts in which the pore volume was not a limiting factor, the high surface area and high acidity are the main factors responsible of the catalytic performance. For better comprehension, the yield of h-GTBE vs structural properties has been plotted in Figure 9.

#### 3.2.3. Reusability and Catalyst Stability

The stability of the solids that exhibited the best results (catalysts **1**, **3** and **6**) was studied during the different cycles of glycerol etherification; the results are shown in Figure 10. 

As shown, the glycerol conversion slightly decreased after each use for all the catalysts, being higher for Ti SiO_2_ PrSO_3_H (**3**). 

In particular, at the 4th cycle of the reaction, the glycerol conversion decreased by around 10% for catalysts **1** and **6** (from 76% to 67% and from 78% to 68%, respectively) and by 14% for catalyst **3** (from 64% to 50%). The leaching of the sulfonic groups in the catalysts was ruled out, since the value of sulfur content in the spent catalysts (after the 4th cycles) was almost the same as that of the fresh materials (Table 1). Therefore, this loss of activity could be associated with a poisoning of the catalyst from organic molecules that could have remained adsorbed on the solid after the reaction, as we previously observed in other catalytic systems [9]. To corroborate this, the spent catalysts (after the 4th cycle) were subjected to an extraction with ethanol under reflux for 3 h. Then, the solids were dried and the liquid fraction was distilled to eliminate the solvent ethanol. The organic fraction was then analyzed by GC, and glycerol and other products of the reaction (mainly MTBGs and DTBGs) were found (Figure 11).

Using this procedure, the catalysts could be restored for their next use in the etherification reaction. In fact, in order to verify this, another cycle (5th reuse) of glycerol etherification was carried out in the presence of all the restored catalysts. As shown in Figure 10, materials **1** and **6** can catalyze a glycerol conversion of 73% and 74% with a yield to h-GTBE of 15% and 16%, respectively. Also catalyst **3**, whose loss of activity was higher, allowed us to achieve a glycerol conversion of 59% and a yield to h-GTBE of 8%. These values are very close to those obtained with fresh catalysts, confirming that the adsorption of the organic fraction in the solid is the main reason for the partial deactivation that these solids underwent. 

## 4. Conclusions

Several sulfonic acid catalysts, i.e., silica, silica-titania and titania-based materials, were tested in the production of high ethers (h-GTBE) by the etherification reaction of glycerol with tert-butyl alcohol under microwave irradiation. Among the studied materials, the catalysts that showed the best performance were sulfonic silica-based derivatives **1** and **6**, with almost total glycerol conversion and 25% yield to h-GTBE in 30 min reaction. By the correlation of the structural properties of the materials to their activity, the performance of the catalysts was shown to be influenced mainly by the surface area and acidity; in particular, catalysts **1** and **6** showed high surface area (616 and 406 m^2^g^−1^, respectively) and high acidity (1.0 and 0.9 mmol H^+^g^−1^). Also, the pore volume played an important role. In fact, the solids with a pore volume equal or superior to 0.07 cm^3^/g showed a linear correlation between acidity and yield to high ethers. All these data indicate that the textural properties are a crucial factor for this reaction, because they affect both the interaction of the reactants with catalytic sites and the diffusion of reactants and products to and from the catalyst structure. Moreover, in terms of recycling the materials, catalysts **1** and **6** showed high stability over five cycles; the activity slightly decreased due to the absorption of the reactants and the products on the supports. By removing the pollutants, the catalysts were easily restored, as highlighted in the 6th cycle. 

Finally, sulfonic silica catalysts with the appropriate textural properties combined with high acidity showed high efficiency in microwave-assisted glycerol etherification, both in terms of glycerol conversion and selectivity to h-GTBE, and in terms of stability over several cycles. It is important to note that in this study, particular attention was paid to the sustainability of the process by using second raw material (glycerol), recycling of the catalysts and an energy saving procedure (microwaves).

## Figures and Tables

**Figure 1 materials-13-01584-f001:**
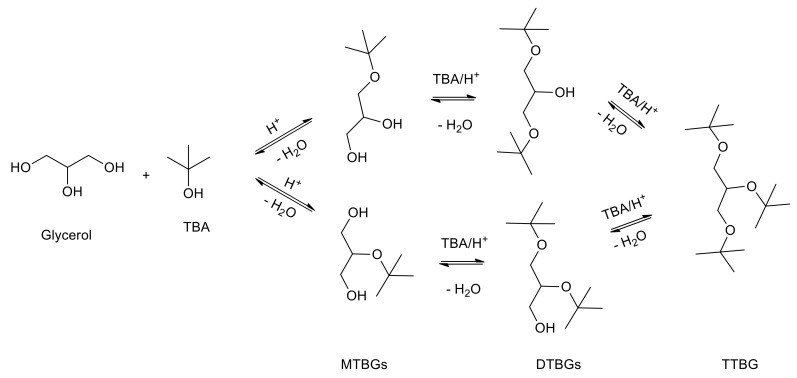
Reaction pathways for the etherification of glycerol with tert-butyl alcohol.

**Figure 2 materials-13-01584-f002:**
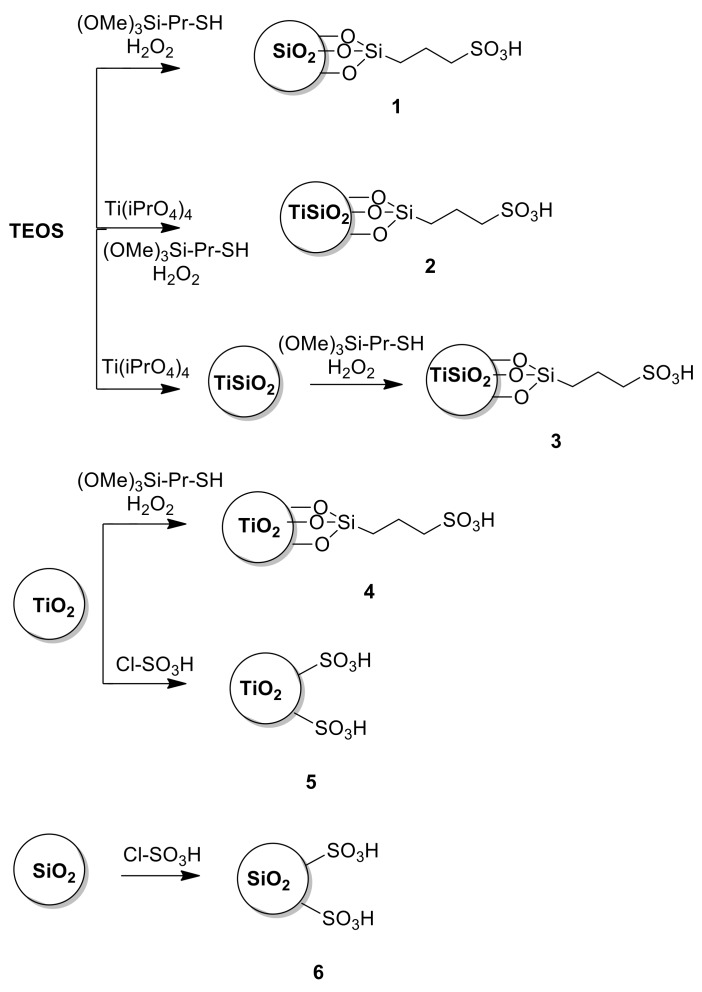
Chemical procedures for the synthesis of sulfonic catalysts.

**Figure 3 materials-13-01584-f003:**
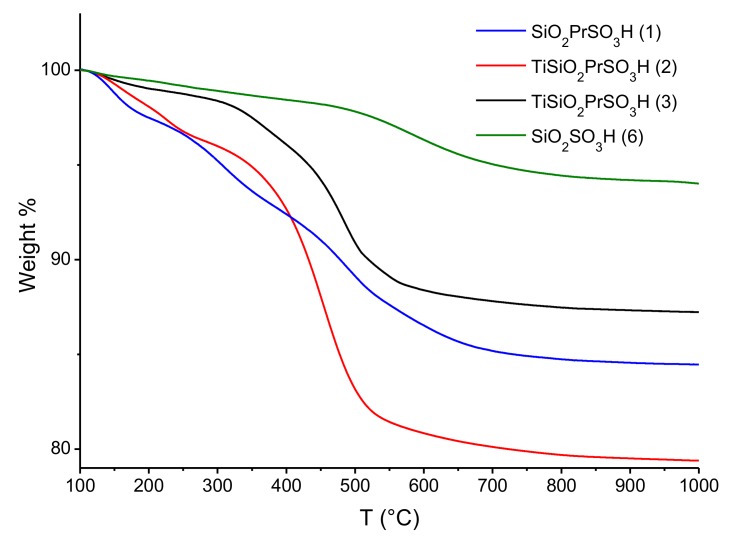
Thermogravimetric curves of sulfonic silica-based derivatives.

**Figure 4 materials-13-01584-f004:**
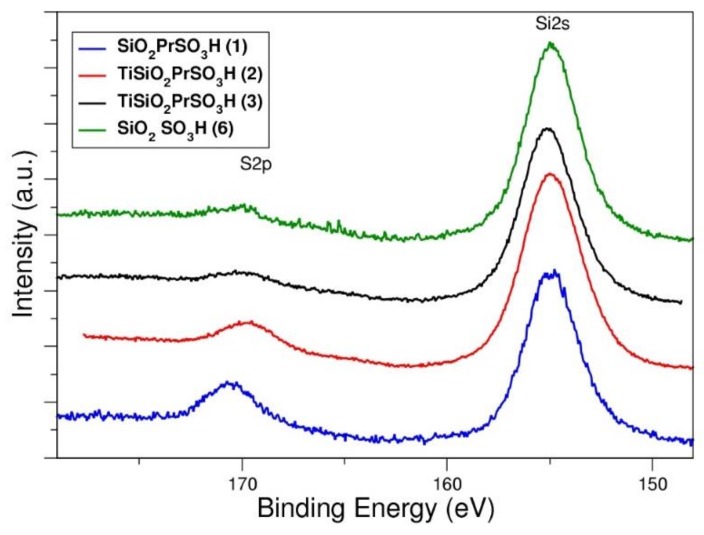
XPS spectra of propylsulfonic silica-based materials (**1, 2, 3** and **6**).

**Figure 5 materials-13-01584-f005:**
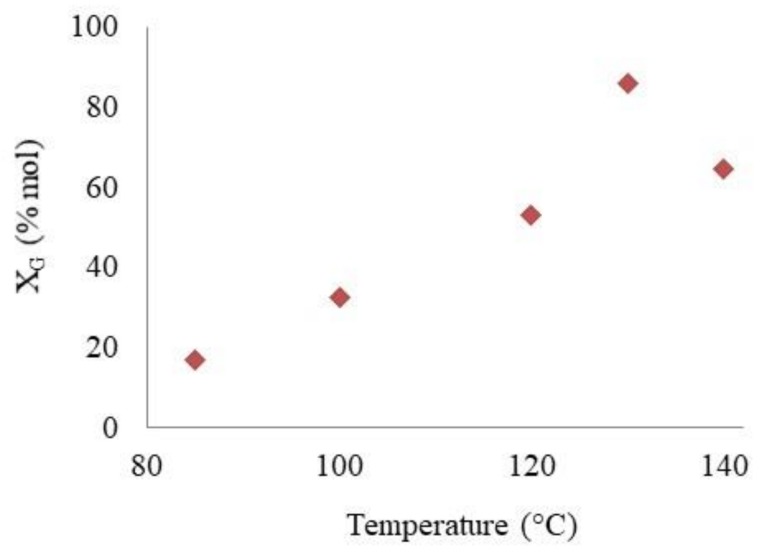
Influence of reaction temperature on glycerol conversion over catalyst **1**. Reaction conditions: TBA/G molar ratio = 4; catalyst **1** amount 5 wt.%; t = 15 min.

**Figure 6 materials-13-01584-f006:**
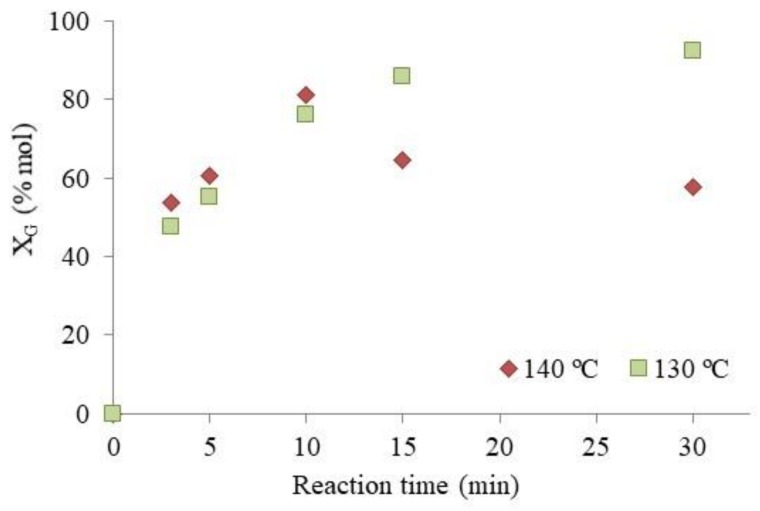
Glycerol conversion vs time profile over catalyst **1** at 130 °C (green squares) and 140 °C (red squares). Reaction conditions: TBA/G molar ratio = 4; catalyst amount 5 wt.%.

**Figure 7 materials-13-01584-f007:**
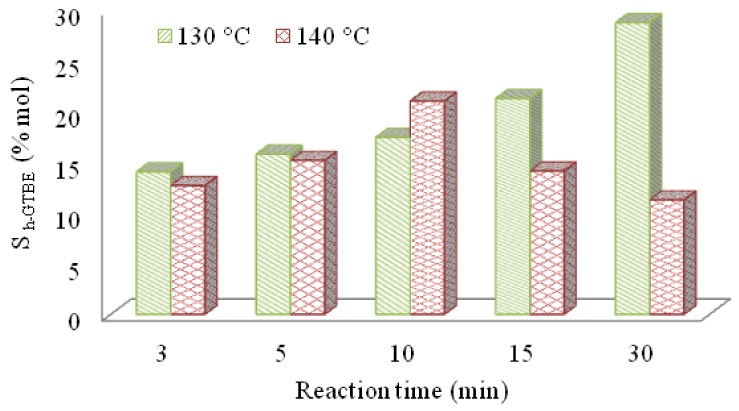
Selectivity to h-GTBE vs time profile at 130 °C (green lines) and 140 °C (red squares) over catalyst **1**. Reaction conditions: TBA/G molar ratio = 4; catalyst amount 5 wt.%.

**Figure 8 materials-13-01584-f008:**
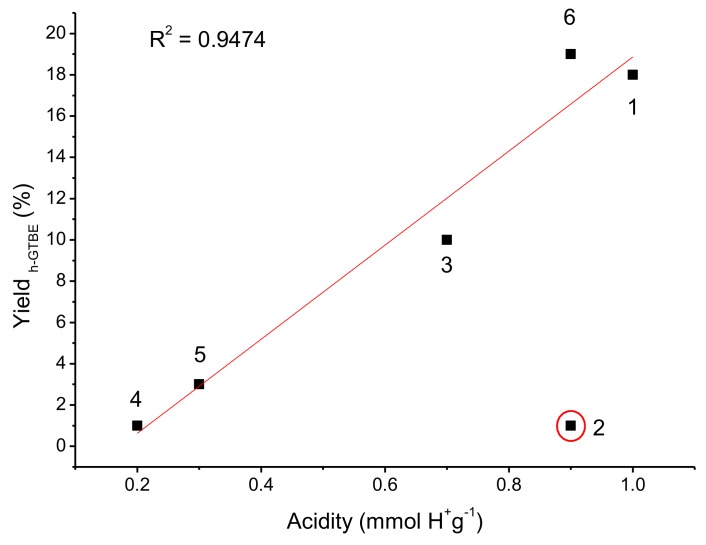
h-GTBE yield vs acidity (mmol H^+^/g) of the sulfonic catalysts **1–6** (T = 130°C; t = 10 min).

**Figure 9 materials-13-01584-f009:**
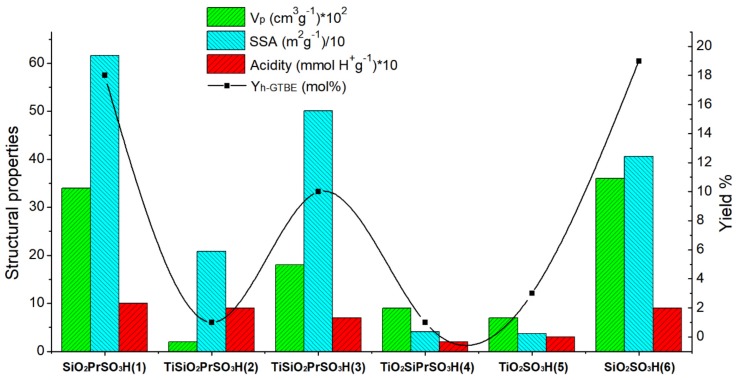
Performance of the catalysts: structural properties and acidity vs yield.

**Figure 10 materials-13-01584-f010:**
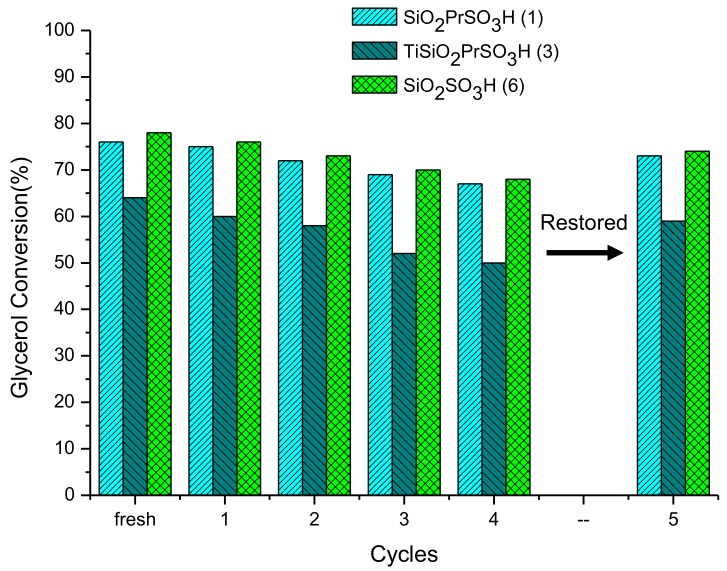
Recycle runs over catalyst **1**, **3** and **6**. Reaction conditions: catalyst amount 5 wt.%, TBA/G ratio = 4, T = 130 °C, t = 10 min.

**Figure 11 materials-13-01584-f011:**
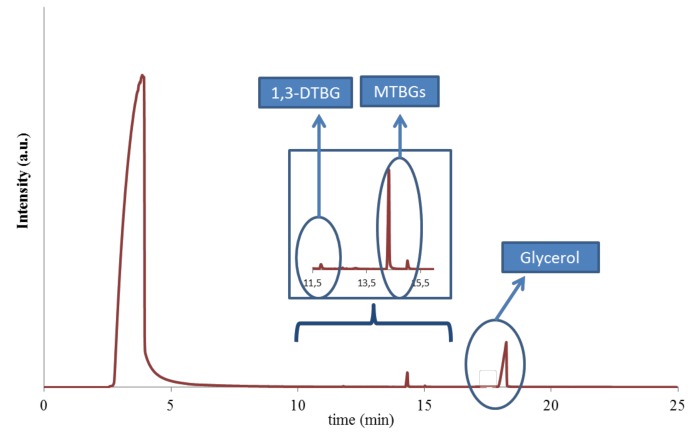
Chromatogram of the organic fraction absorbed in the catalysts.

**Table 1 materials-13-01584-t001:** Textural properties, XPS atomic ratio and acid capacity of the prepared materials.

Catalyst	BET		TGA	XPS			Acidity * (mmol H^+^g^−1^)
	SSA (m^2^g^−1^)	Vp (cm^3^g^−1^)	(% wt)	S/Si	S/Ti	Ti/Si	
SiO_2_ PrSO_3_H (1)	616	0.34	15.5	0.13	-	-	1 (0.9)
Ti SiO_2_ PrSO_3_H (2)	208	0.02	20.6	0.08	-	0.008	0.9
Ti SiO_2_ PrSO_3_H (3)	501	0.18	12.8	0.06	-	0.010	0.7 (0.7)
TiO_2_-SiPrSO_3_H (4)	41	0.09	5.9	0.35	0.19	1.887	0.2
TiO_2_-SO_3_H (5)	37	0.07	2.1	-	0.17	-	0.3
SiO_2_ SO_3_H (6)	406	0.36	6.0	0.04			0.9 (0.8)

* Acidity was calculated by XRF, considering that all the sulfur measured is in sulfonic form, as determined by XPS. The acidity of the spent catalyst after 4 cycles is given in brackets.

**Table 2 materials-13-01584-t002:** Catalytic activity of sulfonic materials **1–6** on glycerol etherification.

Catalyst	X_G_* (%)	S_MTBG_* (%)	S_h-GTBE_* (%)	Y_h-GTBE_* (%)
SiO_2_ PrSO_3_H (1)	76 (93)	76 (73)	24 (27)	18 (25.1)
Ti SiO_2_ PrSO_3_H (2)	14 (16)	94 (92)	6 (8)	1 (1.3)
Ti SiO_2_ PrSO_3_H (3)	64 (78)	86 (83)	14 (17)	10 (13)
TiO_2_-SiPr-SO_3_H (4)	16 (19)	95 (92)	5 (8)	1 (1.5)
TiO_2_-SO_3_H (5)	22 (24)	87 (84)	13 (16)	3 (3.8)
SiO_2_-SO_3_H (6)	78 (93)	76 (73)	24 (27)	19 (26)

Reaction conditions: catalyst amount 5 wt.%, TBA/G ratio = 4, T =130 ° C, t = 10 min. * In brackets, data at 30 min of reaction time.

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
