# Peer review of "Microwave-Assisted Glycerol Etherification Over Sulfonic Acid Catalysts"

_materials, 2020, doi:10.3390/ma13071584_

Round 1

Reviewer 1 Report

The ms reports the study on the synthesis, characterization and catalytic activity of a series of sulfonic acid silica or titania based catalysts. The authors have found the correlation between the structure of the elaborated materials (i.e. surface area, pore volume and acidity) and their activity in the etherification of glycerol with tert-butyl alcohol under microwave irradiation. Moreover, recycling experiments performed over the most active catalytic materials showed maintenance of catalytic behavior during several runs.

The work is quite well presented and supported by multiple experiments. I believe that the manuscript is suitable to be published in catalysts after minor revision. Please, see the comments below.

The authors, actually, have not checked stability of the catalysts and just tested their reusability. However, it’s also necessary to confirm stability of the catalyst structure under the reaction conditions employed. Measure, please, specific surface area and pore volume of reused catalysts to confirm or not stability. Determination of the content of sulfur in the reused catalysts as well as in the reaction mixture after separation of the catalyst would be also useful to confirm (or not) the leaching.

Author Response

Dear Dr Istoan,

Thanks for your e-mail and the enclosed comments of the reviewers. Thanks also the reviewers for the valuable comments that improved the manuscript. I have revised the manuscript “Microwave assisted glycerol etherification over sulfonic acid catalysts” (Manuscript ID: materials-755561), and the comment upon the interesting remarks pointed out by the referees are listed below. In the article, all the variations are written in red, in order to better notice the changes.

Looking forward to hearing from you.

Yours sincerely, 

Dr. Maria Luisa Testa

Reply to Referee 1

COMMENTS AND SUGGESTIONS FOR AUTHORS.

The ms reports the study on the synthesis, characterization and catalytic activity of a series of sulfonic acid silica or titania based catalysts. The authors have found the correlation between the structure of the elaborated materials (i.e. surface area, pore volume and acidity) and their activity in the etherification of glycerol with tert-butyl alcohol under microwave irradiation. Moreover, recycling experiments performed over the most active catalytic materials showed maintenance of catalytic behavior during several runs. The work is quite well presented and supported by multiple experiments. I believe that the manuscript is suitable to be published in catalysts after minor revision. Please, see the comments below:

The authors, actually, have not checked stability of the catalysts and just tested their reusability. However, it’s also necessary to confirm stability of the catalyst structure under the reaction conditions employed. Measure, please, specific surface area and pore volume of reused catalysts to confirm or not stability. Determination of the content of sulfur in the reused catalysts as well as in the reaction mixture after separation of the catalyst would be also useful to confirm (or not) the leaching.

Author (A): We agree with the referee suggestions. The leaching has been disregarded by the measure of the sulfur in the spent catalyst (after 5 cycles). In fact, it was find that after all cycles the acidity related to the amount of sulphur remain the same. This information has been included in Table 1 of the manuscript. As concern other characterization of the spent catalyst, we are agree with the referee but, as you know, due to the extraordinary situation created with the spread of COVID-19 and the situation that we are living now in both countries, Italy and Spain, the characterization that the reviewer suggests is impossible to afford at this moment.

Reviewer 2 Report

Dear Authors,

Well planned research. Very simple reaction but with great application potential thanks to solid-phase catalysis. Tested catalysts well described and characterized. Additionally, optimization of reaction conditions has been done. I have some small comments:
- The etherification reaction procedure is unclear. There is no microwave power used for the reaction. Was the microwave power or temperature set for the experiment (or both?)?
- Abbreviations used in the work should be translated the first time they are used, e.g. FAME, TEOS (so that it is without guesses, especially if you are not in the subject).
- The work contains a few editorial errors.

Author Response

Dear Dr Istoan,

Thanks for your e-mail and the enclosed comments of the reviewers. Thanks also the reviewers for the valuable comments that improved the manuscript. I have revised the manuscript “Microwave assisted glycerol etherification over sulfonic acid catalysts” (Manuscript ID: materials-755561), and the comment upon the interesting remarks pointed out by the referees are listed below. In the article, all the variations are written in red, in order to better notice the changes.

Looking forward to hearing from you.

Yours sincerely,

Dr. Maria Luisa Testa

Reply to Referee 2

COMMENTS AND SUGGESTIONS FOR AUTHORS: Well-planned research. Very simple reaction but with great application potential thanks to solid-phase catalysis. Tested catalysts well described and characterized. Additionally, optimization of reaction conditions has been done. I have some small comments:

Reviewer (R): The etherification reaction procedure is unclear. There is no microwave power used for the reaction. Was the microwave power or temperature set for the experiment (or both?)

Author (A): Yes, the reaction is assisted by the microwaves and in order to clarify this, additional information was included in experimental section 2.2.

R: Abbreviations used in the work should be translated the first time they are used, e.g. FAME, TEOS (so that it is without guesses, especially if you are not in the subject).

A: Regarding the reviewer suggestion, abbreviations was translated the first time that they appeared in the text.

R: The work contains a few editorial errors.

A: The manuscript was carefully revised.

Round 2

Reviewer 1 Report

It's a pity, of course, that the revision of the article coincided with a difficult quarantine period. The authors have made those edits that were able for this period. So I leave it to the Editor to decide whether it is enough to publish as it is or postpone the article until the end of quarantine.